# Diffusion Tensor and Dynamic Contrast-Enhanced Magnetic Resonance Imaging Correlate with Molecular Markers of Inflammation in the Synovium

**DOI:** 10.3390/diagnostics12123041

**Published:** 2022-12-05

**Authors:** Deepak Tripathi, Rishi Awasthi, Vikas Agarwal, Vinita Agrawal, Ram Kishore Singh Rathore, Kusum Sharma, Chandra Mani Pandey, Rakesh Kumar Gupta

**Affiliations:** 1Department of Clinical Immunology and Rheumatology, Sanjay Gandhi Post Graduate Institute of Medical Sciences, Lucknow 226014, UP, India; 2Department of Radiodiagnosis, Sanjay Gandhi Post Graduate Institute of Medical Sciences, Lucknow 226014, UP, India; 3Department of Pathology, Sanjay Gandhi Post Graduate Institute of Medical Sciences, Lucknow 226014, UP, India; 4Department of Mathematics and Statistics, Indian Institute of Technology, Kanpur 208016, UP, India; 5Department of Medical Microbiology, Postgraduate Institute of Medical Education and Research, Chandigarh 160012, PB, India; 6Department of Biostatistics, Sanjay Gandhi Post Graduate Institute of Medical Sciences, Lucknow 226014, UP, India

**Keywords:** rheumatoid arthritis, ankylosing spondylitis, osteoarthritis, tuberculosis, undifferentiated arthritis, synovial histology, markers of inflammation

## Abstract

Objectives: It is difficult to capture the severity of synovial inflammation on imaging. Herein we hypothesize that diffusion tensor imaging (DTI) derived metrics may delineate the aggregation of the inflammatory cells and expression of inflammatory cytokines and dynamic contrast-enhanced (DCE) imaging may provide information regarding vascularity in the inflamed synovium. Patients and methods: Patients with knee arthritis (>3-months duration) underwent conventional (T2-weighted fast spin echo and spin echo T1-weighted images) as well as DTI and DCE MRI and thereafter arthroscopic guided synovial biopsy. DCE and DTI metrics were extracted from the masks of the segments of the inflamed synovium which enhanced on post-contrast T1-weighted MRI. These metrics were correlated with immunohistochemistry (IHC) parameters of inflammation on synovium. Statistical analysis: Pearson’s correlation was performed to study the relationship between DTI- and DCE-derived metrics, IHC parameters, and post-contrast signal intensity. Linear regression model was used to predict the values of IHC parameters using various DTI and DCE derived metrics as predictors. Results: There were 80 patients (52 male) with mean age 39.78 years and mean disease duration 19.82 months. Nineteen patients had tuberculosis and the rest had chronic undifferentiated monoarthritis (*n* = 31), undifferentiated spondyloarthropathy (*n* = 14), rheumatoid arthritis (*n* = 6), osteoarthritis (*n* = 4), reactive arthritis (*n* = 3), ankylosing spondylitis (*n* = 2), and juvenile idiopathic arthritis (*n* = 1). Fractional anisotropy (FA), a metric of DTI, had significant correlation with number of immune cells (r = 0.87, *p* < 0.01) infiltrating into the synovium and cytokines (IL-1β, r = 0.55, *p* < 0.01; TNF-α, r = 0.42, *p* < 0.01) in all patients and also in each group of patients and adhesion molecule expressed on these cells in all patients (CD54, r = 0.51, *p* < 0.01). DCE parameters significantly correlated with CD34 (blood flow, r = 0.78, *p* < 0.01; blood volume, r = 0.76, *p* < 0.01) in each group of patients, a marker of neo-angiogenesis. FA was the best predictor of infiltrating inflammatory cells, adhesion molecule and proinflammatory cytokines. Amongst the DCE parameters, blood volume, was best predictor of CD34. Conclusion: DTI and DCE metrics capture cellular and molecular markers of synovial inflammation in patients with chronic inflammatory arthritis.

## 1. Introduction

Synovium is the primary site of inflammation in various chronic inflammatory arthritides. Chronic inflammation is characterized by the accumulation of various inflammatory cells in the synovium and increased vascularity as a result of neo-angiogenesis [1]. Secretion of proinflammatory cytokines and matrix-degrading enzymes such as matrix metalloproteinases (MMPs) by accumulated inflammatory cells leads to synovial effusion, damage to cartilage, and erosions in the bone [1,2,3].

Presently, the most definitive way to delineate the severity and type of inflammation is through histological examination of the inflamed synovium. However, it is an invasive process, and not all the joints are amenable to synovial biopsy [4]. Moreover, inflammation severity varies from joint to joint and serial assessments are required to evaluate response to therapy. Amongst the non-invasive methods, magnetic resonance imaging (MRI), ultrasonography, positron emission tomography (PET), and bone scintigraphy have been used to assess the inflammation in the joints [5]. Amongst all the techniques available, MRI has the best sensitivity in assessing synovial inflammation. T1-weighted spin-echo sequences acquired early on after administration of intravenous contrast material better differentiate synovial inflammation from the joint effusion [6,7]. Other parameters such as bone marrow edema, synovial membrane volume, and synovial quality moderately correlate with disease activity [4,8,9]. Synovial thickening may be better picked up by fat-suppressed T2 weighted MRI [10] however, its signal may overlap with joint effusion and thus may not be able to differentiate between the two reliably [11]. A relationship between histological synovial inflammation and synovial volume measurement and bone erosion score after one year has been reported [12]. In a recent report, rheumatoid arthritis score magnetic resonance imaging (RAMRIS) was reported to correlate well with synovitis and bone erosion scores. However, the correlation coefficients for synovitis with clinical and serological inflammatory marker CRP varied from 0.4 to 0.36 only [13]. Therefore, these imaging techniques lack a definite correlation with the severity of the synovial inflammation and day-to-day disease activity and thus these may not be appropriate in reflecting a response to therapy. Dynamic contrast-enhanced MRI measures closely mirror the inflammatory activity in the synovium of the joint [14]. However, it is still an experimental tool and there is no universal consensus about the protocol.

Diffusion tensor imaging (DTI) is a non-invasive imaging technique that measures the diffusion of water molecules in vivo and provides microstructural information about the tissue [15,16]. Recently, it has been shown that diffusion tensor imaging (DTI) derived metrics such as fractional anisotropy (FA) and mean diffusivity (MD) correlate with the expression of proinflammatory molecules such as TNF-α and IL-1β and adhesion molecules such as ICAM-1 on the various inflammatory cells in the brain abscess [17]. Building on this information, in our previous pilot study, we demonstrated that DTI-derived metrics correlated well with the level of proinflammatory cytokines in the synovial fluid [18]. However, the limitation of our study was that DTI metrics signals were generated in the synovium and the levels of proinflammatory cytokines were measured in the synovial fluid which may not go hand in hand. Therefore, in the present study, we aimed to study a direct correlation between the degree of inflammation in the synovium and DTI-derived metrics.

Diffusion anisotropy can be more completely modeled using three basic metrics; linear anisotropy (CL), when diffusion is mainly in the direction corresponding to the largest eigenvalues, planar anisotropy (CP), where diffusion is restricted to a plane spanned by the two eigenvectors corresponding to the two largest eigenvalues, spherical anisotropy (CS) when diffusion of water molecules is unrestricted or isotropic diffusion. Its utility has been explored in a number of clinical studies [16,19,20].

T1-weighted dynamic contrast-enhanced (DCE) MRI using simple analyses of changes in the signal enhancement curve, such as measuring the maximum enhancement and the rate of initial enhancement, had been used to quantify the extent of synovial inflammation and response to treatment [21]. Quantitative DCE-MRI based on pharmacokinetic modeling has been applied in a number of studies on various pathologies of different body regions including the knees of children with juvenile idiopathic arthritis [22]. The use of appropriate mathematical models can allow the quantitative assessment of synovial inflammation and disease activity in adults with RA by looking at hemodynamic and pharmacokinetic changes in the affected regions through DCE-MRI [23]. Moreover, since neo-angiogenesis is an important component of chronic inflammation, we have used perfusion MRI to evaluate vascularity in the inflamed synovium [24]. Herein we hypothesize that DTI-derived metrics may provide information about the aggregation of the inflammatory cells and perfusion imaging may provide information regarding neo-angiogenesis in the inflamed synovium. The aim of the present study was to correlate DTI and DCE-MRI-derived metrics with molecular markers of inflammation in the synovium of the knee joint in patients with chronic inflammatory and infective arthritis.

## 2. Materials and Methods

### 2.1. Patients

Newly diagnosed subjects, age >18 years with inflammatory monoarthritis of the knee of >3 months duration with mono/oligo-articular flare of the knee or to rule out infection in an already inflamed joint were enrolled in the study. “Flare” was defined as either involvement of the new joint or exacerbation of inflammation in the previously inflamed joint. The study was approved by the institutional review board and written informed consent was obtained from all study subjects. The study was conducted according to the guidelines of the Declaration of Helsinki and approved by the institute ethics committee of the Sanjay Gandhi Postgraduate Institute of Medical Sciences, Lucknow, approval number 2011-19-SRF-58, dated 21 March 2012. The involved knee joints were subjected to MRI and DTI followed by arthroscopic guided biopsy of the synovium and aspiration of the synovial fluid (SF) from the same joint within 48 h of the MRI.

### 2.2. Magnetic Resonance Imaging

Conventional MRI including DTI and DCE-MRI was performed using a standard 8-channel knee coil on a 3-T MR scanner (SignaHDxt, GE Healthcare, Milwaukee). Conventional T2-weighted fast spin echo (FSE) images and spin echo (SE) T1-weighted images were acquired. T2-weighted images were acquired with following parameters; repetition time (TR) (ms)/echo time (TE) (ms)/echo train length (ETL)/no. of excitations (NEX) = 6000/85/16/4. T1 weighted images were acquired with TR/TE/NEX = 1000/14/2. Fat-saturated T1- and T2-weighted images were acquired from adjacent (overlaid), 3 mm thick axial sections with 240 × 240 mm field of view (FOV) and image matrix of 256 × 256.

DTI data were acquired using a single-shot echo-planar dual SE sequence with ramp sampling. The b-factor was set to 1000 s/mm^2^. The acquisition parameters were: FOV = 240 × 240 mm, TR~8 s, TE~100 ms, and number of excitations = 8. The acquisition matrix was 128 × 80. K-space data (128 × 128 and zero-filled to generate an image matrix of 256 × 256) was constructed using homodyne algorithm. The DTI data were processed as described in detail elsewhere [18].

DCE-MRI was performed using a three-dimensional spoiled gradient recalled echo (3D-SPGR) sequence [TR/TE/flip-angle/NEX/slice-thickness/FOV/matrix size = 5.1 ms/2.1 ms/10°/0.7/3 mm/240 × 240 mm/256 × 256 mm, number of phases = 40]. Gadolinium contrast (Gd-DTPA-BMA; Omniscan, GE Healthcare, Chicago, IL, USA) was administered intravenously through a power injector (Malinkrodt Optistar LE) at 5 mL/s and a dose of 0.2 mmol/kg body weight, followed by 30 mL saline flush, at the start of the sixth acquisition. Serial 1120 images (temporal resolution: 7.85 s) were acquired over 40 time points for 28 slices. Prior to 3D SPGR, T1-weighted, FSE, and fast double spin-echo proton density (PD) weighted and T2-weighted imaging were performed to quantify voxel-wise the pre-contrast tissue longitudinal relaxation time T_10_ as described previously [25]. Finally, a post-contrast T1-weighted FSE (fat saturation) was also acquired following the SPGR with same imaging parameters as pre-contrast one. All imaging was acquired with same slice position and thickness in order to register all the datasets.

### 2.3. Data Quantification

Masks of synovial regions that enhanced on post-contrast T1-weighted imaging were created using an automated segmentation algorithm as described [26]. Created masks were used to segment the inflamed synovium to extract various DCE (BV, BF, k_ep_) and DTI (FA, MD, CL, CP, CS) metrics from the segmented regions (Figure 1). Two observers RA and DT blinded to the diagnosis performed the image quantification analysis.

### 2.4. Arthroscopic Biopsy and Histopathology

Arthroscopic needle biopsy was carried out as follows: after disinfection and local anesthesia of the skin and joint, a 1.8–2.7 mm needle arthroscope (Olympus) was introduced into the knee. Synovial fluid was aspirated, and lavage of the joint cavity was performed to allow macroscopic evaluation of hyperemia and hypertrophy of the synovial membrane. Biopsies were taken at inflamed sites under direct visualization, followed by another lavage to remove blood and debris. A total of 8–10 biopsy specimens were obtained from the suprapatellar anterior, medial, and lateral compartments.

### 2.5. Histopathology and Synovitis Severity Grading

The grading of the synovial membranes was carried out on routine hematoxylin and eosin (H&E)-stained slides. The histopathological findings were compared using the histological scoring system reported by Rooney, which consists of six parameters: degree of synovial hyperplasia, fibrosis, number of blood vessels, perivascular lymphocyte infiltration, focal aggregates of lymphocytes, and diffuse infiltrates of lymphocytes [27]. Patients were classified according to varying degrees of synovitis such as mild (0.0–1.5), moderate (1.6–3.0), and severe (3.1–5.0) (Figure 2).

### 2.6. Immunohistochemistry and Microscopic Analysis

The infiltration by various types of inflammatory cells, vascularity, and cytokine expression in the synovium was analyzed by immunohistochemistry using antibodies CD3, (Dako, Glostrup, Denmark) 1:200, CD4, (Dako, Glostrup, Denmark) 1:50; CD8, (Dako, Glostrup, Denmark) Ready to use; CD20 (Dako, Glostrup, Denmark) 1:200; CD34 (Dako, Glostrup, Denmark) 1:50; CD54 (Leica) 1:25; CD68 (Dako, Glostrup, Denmark) 1:200; CD138 (Dako, Glostrup, Denmark), ready to use. TNFα R (Santa Cruz Biotechnology, Inc., Santa Cruz, CA, USA), 1:100; IL-1β, (Santa Cruz Biotechnology, Inc., Santa Cruz, CA, USA) 1:100. Immunohistochemistry was performed on formalin-fixed, paraffin-embedded sections. A polymer-based peroxidase method with diaminobenzidine (Novocastra Laboratories, Leica microsystems, London, UK) as chromogen was used to detect the bound antibodies.

The sections were evaluated under a light microscope (Olympus DP70). Only nucleated cells with clear-cut cytoplasmic or surface staining were counted and three hot spots; areas of synovium containing highest number of cells, per section were identified. Cells that were positive for IHC markers were counted in 10 consecutive high-power fields (X400) per hot spot. Thereafter, an average number of positive cells per three hot spots was calculated and results were expressed as total number of positive cells per square millimeter [28,29]. Two observers VA and DT blinded to the diagnosis performed the immunohistological analysis.

### 2.7. Quantitation of Inflammatory Cytokines from Synovial Fluid

The human soluble IL-6, IL-1β, and TNF-α in synovial fluid were quantitatively measured by enzyme-linked immunosorbent assay (ELISA) (BD opt EIA, San Diego, CA, USA) in duplicate. ELISA for these cytokines was performed as per manufacturer’s guidelines. Number of inflammatory cytokines in samples was determined by standard curve.

### 2.8. Statistical Analysis

Spearman correlation with Bonferroni correction (*p* < 0.0055) was performed to study the relationship between different DTI- and DCE-derived metrics, IHC parameters, inflammatory cytokines measured in SF, and post-contrast signal intensity. Linear regression model was used to predict the values of IHC parameters of inflammation by using various DTI and DCE-derived matrices as predictors.

Model: y = β_0_ + β_1_x where y is estimated value of IHC parameters and x is determinant (DTI or DCE parameter). β_0_ is constant term and β_1_ is the coefficient of determinant parameter. Multivariate linear regression was used to estimate the best DTI and DCE parameters to predict synovial inflammation based on Rooney score.

All the statistical computations were performed using the SPSS (Statistical Package for Social Sciences; version 12.0, SPSS Inc., Chicago, IL, USA) statistical software.

### 2.9. Sample Size Estimation

Sample size estimation was based on our previous study results where correlation coefficients between DTI parameters and synovial fluid pro-inflammatory cytokines varied between 0.4 to 0.6 [18]. For the present study, the value of correlation coefficient between various DTI parameters was assumed to be 0.4 to be significant at *p* < 0.05 and 95% power. The minimum sample size required for the study under these assumptions was 75. Considering few drop outs it was planned to take 83 subjects for the study.

## 3. Results

There were 80 patients, mean age of 39.78 years (range 18–76) with 52 being male (Table 1). The mean disease duration was 19.82 months. Nineteen patients had tuberculosis (histology, culture, and PCR positive) and the rest had undifferentiated spondyloarthropathy (*n* = 14), chronic undifferentiated monoarthritis (*n* = 31), rheumatoid arthritis (*n* = 6), osteoarthritis (*n* = 4), ankylosing spondylitis (*n* = 2), reactive arthritis (*n* = 2) and juvenile idiopathic arthritis (*n* = 1). All six patients with RA were receiving methotrexate (mean dose 17.5 mg/week) and NSAIDs. None of them were on oral or parenteral glucocorticoids or biological therapy. Rests of the patients were receiving NSAIDs only, at the time of inclusion in the study.

Inflammatory arthritis patients were classified in to mild, moderate, and severe disease on the basis of Rooney synovitis score (Table 2). Correlation between DTI and DCE parameters with Rooney synovitis score revealed that FA, MD, CL, BF, BV, and k_ep_ significantly correlated with severe synovial inflammation (Rooney synovitis score 3.1–5) whereas FA only correlated with moderate synovial inflammation (Rooney synovitis score 1.6–3) (Appendix A). The mean values of various DTI and DCE parameters and IHC and ELISA parameters for different group of patients are presented in (Table 3). Amongst the DTI parameters, FA significantly correlated with all the inflammatory cells infiltrating into the synovium in all patients (Table 4, Figure 3) and in different groups of patients also (Appendix A) and various proinflammatory cytokines (TNF-α and IL1β) and adhesion molecule (CD 54) expressed on these cells. This correlation was highly significant for total number of cells infiltrating into the synovium. In contrast to FA, MD had a significant negative correlation with infiltrating cells and proinflammatory cytokines. CL had a significant correlation with inflammatory cells and proinflammatory cytokines and the total number of cells infiltrating the synovium. On the other hand, D, KepCE parameters significantly correlated with CD34, a marker of angiogenesis. Similarly, DTI parameters, FA and MD correlated with synovial fluid cytokines IL-1β and TNF-α (Appendix A).

In the synovial fluid, DTI parameters; FA, MD, and CL significantly correlated with the synovial fluid proinflammatory cytokines (Table 5) in all patients and in different groups also. In contrast to DTI and DCE parameters, post-contrast intensity (parameter of conventional MRI) did not correlate with any of the IHC parameters of inflammation either in the synovium or synovial fluid (except SF IL-1β).

## 4. Predicting the Values of IHC Parameters in the Synovium of Arthritis Patients Using Various DTI and DCE Metrics

On application of the linear regression model, it was observed that amongst all the DTI parameters, FA was the best predictor of infiltrating T cells (CD3, CD4, CD8), B cells (CD20, CD138), macrophages (CD68), adhesion molecule (CD54), the total number of cells infiltrating into the synovium and proinflammatory cytokines (TNF-α and IL-1β) (Table 6a–c). Other parameters of DTI; MD, CL, and CS were also able to predict various inflammatory cells and cytokines, but variably. However, FA, CP, and k ep predicted the total number of cells infiltrating the synovium (Appendix A), and BV and BF predicted the neo-angiogenesis (CD 34) in synovium with the highest percentage of accuracy. Amongst the DCE parameters, blood volume, blood flow, and k_ep_ were able to predict CD34 (Table 7 and Appendix A).

## 5. Discussion

The results of the present study showed a significant correlation between DTI-derived metrics with IHC parameters of proinflammatory cellular aggregates, adhesion molecule, and proinflammatory cytokines and DCE parameters with markers of neo-angiogenesis, respectively. Thereby, almost all the components of chronic synovial inflammation are captured on imaging with high accuracy.

On application of a high-power magnetic field to water hydrogen and hydroxy ions are created with a particular line-up according to the magnetic field. When the magnet is switched off, these ions are free to move (diffusion) in any direction. Fractional anisotropy (FA) measures interference in the movement of these ions due to the vectrality of the tissue structure and creates anisotropy or fractional anisotropy (FA). Its values vary between 0 (equal probability of moving in all directions) or isotropic diffusion and a maximum value of 1, where movement is possible in only one direction true for thin fibers (anisotropic structures). White matter fibers in the brain and spinal cord are anisotropic structures and have high FA values (but less than 1). In the normal synovium, there is not much vectrality however, in the inflamed synovium it was reported to be high as compared to a healthy individual’s synovium. It was also reported that high FA values in the inflamed synovium correlated with synovial fluid TNF-α and IL-1β levels [18]. Earlier, high FA values observed in the brain abscess cavity have been postulated to be due to aggregation of the inflammatory cells [17]. We have observed a strong correlation between FA and various inflammatory cells (CD3, CD4, CD8, CD68), the total number of inflammatory cells, proinflammatory cytokines (TNF-α, IL-1β), and adhesion molecule (CD54) expression on these inflammatory cells. On the basis of this observation, we propose that increased expression of proinflammatory cytokines led to a migration of the inflammatory cells from the vascular compartment to the synovium, and over-expression of adhesion molecule on these inflammatory cells led to the oriented aggregation of the inflammatory cells and thus resulted in increased FA values on DTI MRI.

Mean diffusivity (MD) is the measurement of free diffusivity of the water molecules and is generally inversely related to FA [30,31,32]. A negative correlation between MD and cellular infiltration and proinflammatory cytokines has resulted due to the inability of the water molecules to diffuse freely due to organized inflammatory exudates on the inflamed synovial membrane [18]. This further strengthened our hypothesis that synovial cellular aggregates are responsible for high FA and low MD.

Both CL and CP values are responsible for increased FA values; however, their relative values indicate the shape of the anisotropy [32]. The brain abscess wall has increased CL in comparison to the abscess cavity. Concentric layers of collagen fibers in the brain abscess wall allow water molecules to diffuse in a parallel direction to the collagen fibers only [33] and thus increased CL. Similarly, collagen-rich tissues preferentially have higher rates of diffusion of water molecules along the primary orientation (parallel) of collagen fibers compared to the perpendicular direction [34]. Normal white matter shows high CL in comparison to abscess cavities due to organized neuronal fiber bundles [35]. The increase in CP values has also been explained on the basis of adherent neuroinflammatory molecules that probably imitate the presence of haphazardly arranged vectors and give the more planer model of diffusion tensor in the abscess cavity [17]. In the present study, we observed a significant correlation between linear and planar anisotropy (CL and CP) with the inflammatory cell infiltration (CD3, CD4, CD8, and CD68), the total number of inflammatory cells, and proinflammatory cytokines (TNF-α and IL-1β) expression on these cells suggesting that the increased cellular infiltration along the synovial membrane may have resulted in the increase in both CL and CP in tuberculous synovitis (TS) as compared to non-tubercular inflammatory synovitis (non-TS) cases. On the other hand, the observed inverse correlation of CS with cellular infiltration, proinflammatory cytokines, and adhesion molecule expression in the present study may be attributed to the fact that the increased cell density due to the accumulation of inflammatory cells, resulted in hindrance to free diffusion.

Inflammation is characterized by cellular infiltration and increased vascularity [36,37]. In acute settings, vascularity is increased by vasodilatation in response to various inflammatory signals, however, in chronic inflammation vascularity is increased by neo-angiogenesis. For the assessment of vascularity in the chronically inflamed synovium, we have used DCE. CD34 is a cell surface glycoprotein and is expressed on hematopoietic cells, mesenchymal stem cells, endothelial progenitor cells, endothelial cells of blood vessels, mast cells, dendritic cells, and some soft tissue tumors. DCE parameters (BV, BF) correlated well with the markers of angiogenesis, i.e., CD34, and with the cellular infiltration, proinflammatory cytokines expression, and adhesion molecule on inflammatory cells.

Infections are likely to have more intense inflammation as compared to chronic inflammatory conditions [38,39,40] and therefore are likely to generate higher abnormal DTI and DCE metrics. In fifteen cases that had tubercular synovitis in the present study, higher abnormal DTI and DCE metrics were observed as compared to non-infective chronic inflammatory synovitis in the rest of the cases. Not only higher signals in DTI and DCE metrics were noted, but higher cellular infiltrations, proinflammatory cytokines, and adhesion molecule expression and neo-angiogenesis were also observed on IHC. This observation again signifies that DTI and DCE metrics capture cellular and molecular markers of inflammation and correlated with the degree of inflammation.

Our study is limited by the small number of subjects in each subgroup as per Rooney synovitis score and hence correlation in each subgroup with DTI and DCE parameters may be underpowered and not robust.

To the best of our knowledge, this is the first study that shows a direct correlation between cellular and molecular markers of synovial inflammation. The variability in the DTI and DCE metrics with the intensity of synovial inflammation has the potential to replace synovial histology in the future. It may become a non-invasive tool to assess the degree of synovial inflammation in response to various newer therapeutic molecules across various rheumatic diseases.

## Figures and Tables

**Figure 1 diagnostics-12-03041-f001:**
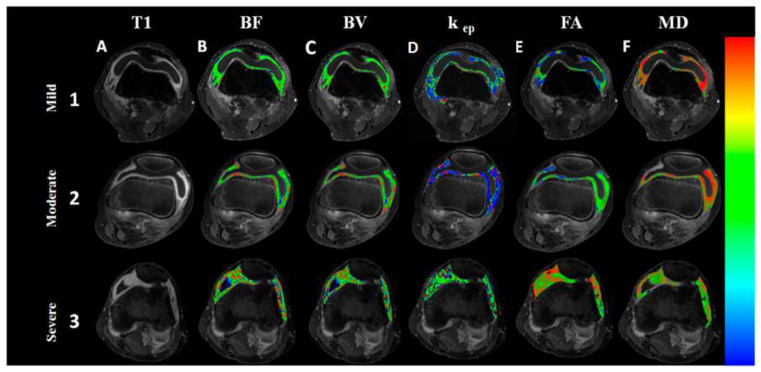
Imaging of the knee joints of arthritis patients presenting with varying grades of inflammation. Imaging of the knee joints of arthritis patients presenting with varying grades of inflammation. Fat-suppressed post-contrast T1-weighted images show increase in contrast enhancement on moving from mild to severe inflammation (lane **A**, 1–3). On post-processing of dynamic contrast-enhanced and diffusion tensor-magnetic resonance imaging data, blood flow (BF), (**B**), blood volume (BV), (**C**), rate transfer coefficient k_ep_ (min^−1^) (**D**) and fractional anisotropy (FA) (**E**) maps show increase, while mean diffusivity (MD) (**F**) map shows decrease in its values on moving from mild to severe synovitis (1–3).

**Figure 2 diagnostics-12-03041-f002:**
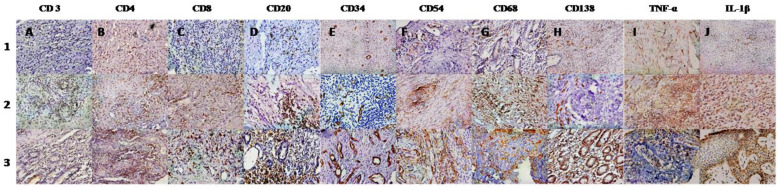
Immunohistochemistry of knee synovium from arthritis patients showing varying grades (1–3) of severity of inflammation and infiltration by various inflammatory cells. Immunohistochemistry of knee synovium from arthritis patients (1, 2, and 3) showing varying grades of infiltration by various inflammatory cells-. T cells (CD3, CD4, CD8) (lane **A**, **B**, **C** respectively); B cells (CD 20) (lane **D**), plasma cells (CD138) (lane **H**); and macrophages (CD 68) (lane **G**). Cells expressing adhesion molecule ICAM-1(CD54) (lane **F**), TNF-α (lane **I**) and IL-1β (lane **J**) are also shown. Immunostaining for endothelial cells (CD34) (lane E) shows the varying density of micro vessels in the synovial tissue.

**Figure 3 diagnostics-12-03041-f003:**
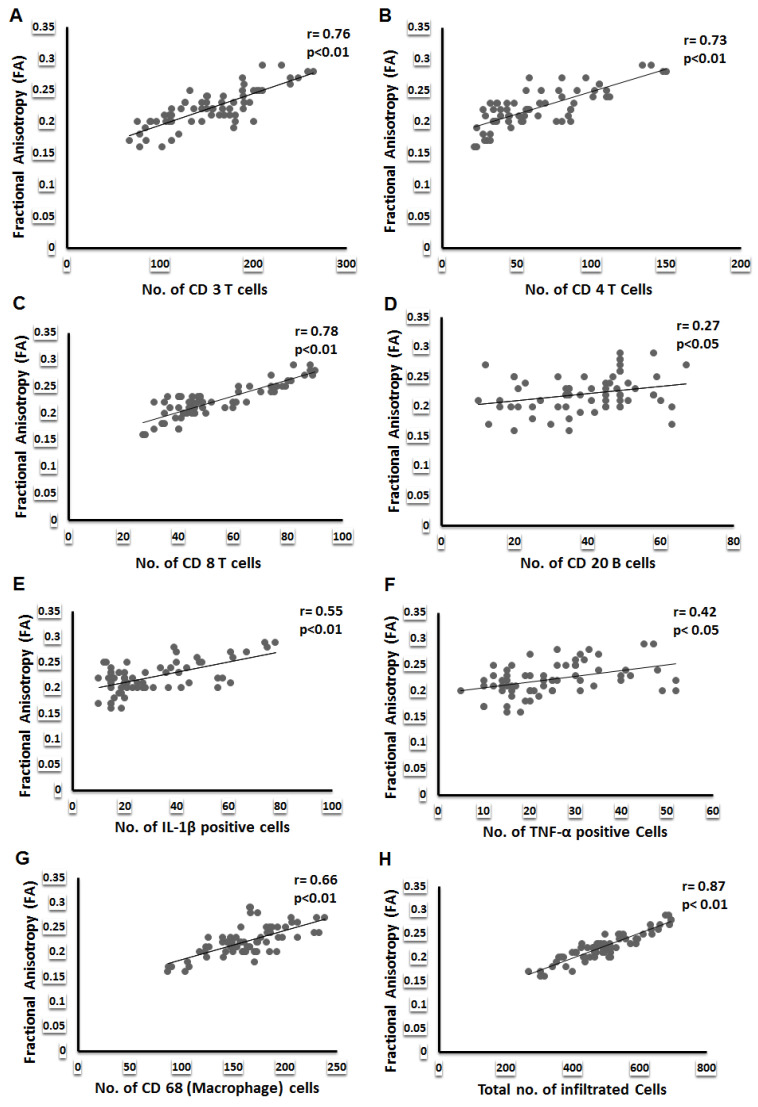
Correlation of inflammatory cell infiltration with fractional anisotropy. Spearman’s correlation scatter plots represent the correlation between (**A**) FA and CD3, (**B**) CD4, (**C**) CD8, (**D**) CD20, (**E**) IL-1β positive cells, (**F**) TNF-α positive cells, (**G**) CD68 and (**H**) total number of inflammatory cells in the synovium.

**Table 1 diagnostics-12-03041-t001:** Clinical and demographic features of patients.

	Total Arthritis Patients (*n* = 80)	Different Groups of Arthritis Patients					
		RA (*n* = 6)	Infective Arthritis (*n* = 19)	USpA (*n* = 14)	UCMA (*n* = 31)	OA (*n* = 4)	Others (*n* = 6)
Age (Mean ± SD)	39.78 ± 12.33	41.67 ± 6.77	39.16 ± 15.18	38.43 ± 13.11	41.52 ± 11.50	48.25 ± 4.85	38.33 ± 5.35
M/F	52/28	2/4	14/5	11/3	18/13	3/1	4/2
Disease Duration (Mean ± SD)	19.82 ± 14.22	17.66 ± 7.63	15.36 ± 10.50	20.85 ± 18.12	23.67 ± 16.46	15 ± 3.46	17.00 ± 7.01
ESR (mm/h) (Mean ± SD)	65.78 ± 30.98	36.50 ± 26.16	52.00 ± 25.07	62.20 ± 19.97	80.34 ± 29.84	32.00 ± 14.14	97.33 ± 17.21
CRP (mg/dl) (Mean ± SD)/Median	3.52 ± 6.86 (0.32–32.70)	4.85 ± 5.43 /(1.01–8.70)	1.56 ± 1.48 /(0.23–4.95)	6.00 ± 10.95 /(0.32–34.50)	2.57 ± 4.05 /(0.32–12.40)	0.96 ± 0.63 /(0.41–1.61)	1.60 ± 1.31 /(0.67–2.53)

RA: rheumatoid arthritis, USpA: undifferentiated spondylarthropathy, UCMA: undifferentiated chronic monoarthritis. OA: osteoarthritis, ESR: erythrocyte sedimentation rate, CRP: C-reactive protein.

**Table 2 diagnostics-12-03041-t002:** Inflammatory arthritis patients classified on the basis of severity of synovitis.

Severity of Synovitis (Rooney Synovitis Score)	Knees (N)
Mild (0.0–1.5)	16
Moderate (1.6–3.0)	27
Severe (3.0–5.0)	37

**Table 3 diagnostics-12-03041-t003:** Descriptive values of imaging and immunohistochemical parameters in different groups of arthritis patients.

DTI Indices (Mean ± SD)	Arthritis Patients (*n* = 80)	Different Groups of Arthritis Patients					
		RA (*n* = 6)	Infective Arthritis (*n* = 19)	USpA (*n* = 14)	UCMA (*n* = 31)	OA (*n* = 4)	Others (*n* = 6)
FA	0.22 ± 0.03	0.21 ± 0.02	0.25 ± 0.02	0.21 ± 0.02	0.21 ± 0.01	0.18 ± 0.01	0.22 ± 0.01
MD (10^−3^ mm^2^s^−1^)	1.62 ± 0.50	1.80 ± 0.28	1.28 ± 0.42	1.65 ± 0.66	1.78 ± 0.43	1.56 ± 0.25	1.88 ± 0.44
CL	0.05 ± 0.02	0.04 ± 0.01	0.08 ± 0.02	0.04 ± 0.01	0.05 ± 0.02	0.07 ± 0.02	0.05 ± 0.03
CP	0.14 ± 0.05	0.15 ± 0.06	0.16 ± 0.03	0.14 ± 0.05	0.14 ± 0.05	0.09 ± 0.06	0.16 ± 0.05
CS	0.5 ± 0.02	0.74 ± 0.01	0.73 ± 0.02	0.75 ± 0.01	0.75 ± 0.02	0.76 ± 0.01	0.75 ± 0.02
**DCE indices**							
BF (mL/100 gm/min)	109.91 ± 42.83	78.90 ± 42.66	150.46 ± 35.42	106.97 ± 39.95	92.38 ± 26.95	72.99 ± 10.15	125.22 ± 49.16
BV (mL/100 gm)	9.54 ± 4.20	8.28 ± 3.36	12.50 ± 4.77	9.32 ± 4.15	8.24 ± 2.88	6.45 ± 2.65	9.85 ± 5.16
k_ep_ (min^−1^)	2.49 ± 1.00	2.27 ± 0.24	3.74 ± 0.83	2.27 ± 0.97	1.91 ± 0.50	1.94 ± 0.64	2.12 ± 0.42
PCI	1820.44 ± 211.64	1875.5 ± 153.3	1852.78 ± 183.33	1713.42 ± 240.11	1874.22 ± 211.84	1801.87 ± 245.50	1750 ± 213.81
**Immune cells in synovium (mean no. of cells/hpf)**							
CD3	154.93 ± 48.65	154.66 ± 38.38	210.75 ± 34.15	135.14 ± 41.87	142.25 ± 32.78	82.25 ± 14.88	141.00 ± 22.90
CD4	63.41 ± 32.85	60.83 ± 23.70	104.31 ± 28.34	48.14 ± 24.56	52.65 ± 18.20	28.00 ± 7.16	49.80 ± 20.11
CD8	53.58 ± 17.62	48.50 ± 17.81	77.31 ± 11.86	44.07 ± 12.90	47.15 ± 7.42	35.50 ± 5.44	50.60 ± 8.32
CD20	39.33 ± 13.96	40.00 ± 15.12	44.56 ± 16.58	35.78 ± 13.90	39.45 ± 12.27	26.75 ± 12.84	41.40 ± 4.09
CD34	52.93 ± 17.27	40.33 ± 16.00	64.50 ± 17.18	49.28 ± 18.75	48.90 ± 14.05	50.25 ± 11.08	59.60 ± 14.55
CD54	36.35 ± 13.14	37.33 ± 16.05	44.62 ± 7.14	34.50 ± 10.63	30.90 ± 11.32	26.25 ± 12.81	43.80 ± 23.27
CD68	163.20 ± 34.61	158.33 ± 10.93	197.62 ± 25.60	158.21 ± 37.74	153.25 ± 23.43	111.50 ± 27.04	154.00 ± 25.34
CD138	36.06 ± 14.49	34.00 ± 11.61	43.43 ± 15.82	31.78 ± 14.78	39.60 ± 12.26	24.25 ± 10.65	22.20 ± 3.49
Total Cells	489.66 ± 106.38	464.66 ± 67.57	629.50 ± 56.20	446.14 ± 82.31	456.35 ± 55.83	321.25 ± 44.67	462.00 ± 67.53
IL-1β	31.09 ± 18.14	19.16 ± 7.02	51.62 ± 15.90	20.85 ± 9.23	30.40 ± 16.22	16.75 ± 5.56	22.60 ± 12.09
TNF-α	24.70 ± 11.51	15.00 ± 3.79	31.99 ± 9.40	17.92 ± 9.06	28.55 ± 12.56	17.50 ± 5.25	22.40 ± 9.44
**Cytokines in the Synovial fluid (pg/mL)**							
IL-6	9788.30 ± 474.44	8200.00 ± 373.01	10,763.75 ± 339.54	11,551.42 ± 485.83	9323.00 ± 593.67	8163.73 ± 510.23	6928.00 ± 398.16
IL-1β	25.46 ± 22.56	24.60 ± 18.80	37.77 ± 26.16	24.33 ± 2.34	17.18 ± 13.05	18.10 ± 4.65	29.30 ± 7.53
TNF-α	88.49 ± 57.25	46.29 ± 34.23	110.91 ± 59.55	126.10 ± 49.80	57.12 ± 36.10	74.56 ± 52.50	96.26 ± 79.43

DTI: diffusion tensor imaging, DCE-MRI: dynamic contrast-enhanced magnetic resonance imaging, FA: fractional anisotropy, MD: mean diffusivity, CL: linear anisotropy, CP: planar anisotropy, CS: spherical isotropy, BV: blood volume, BF: blood flow, k ep: volume transfer constant, PCI: post-contrast signal intensity. IL-1β: interleukin 1β, TNF-α: tumor necrosis factor α, IL-6: interleukin.

**Table 4 diagnostics-12-03041-t004:** Correlation between the values of DTI and MRI indices with various infiltrating immune cells in the synovium (*n* = 80). Results are expressed as R values.

DTI-MRI Indices	Infiltrated Immune Cells in Synovium										
	CD3	CD 4	CD 8	CD 20	CD 34	CD 68	CD 138	CD 54	TNF-α	IL-1β	Total Inflammatory Cells
FA	0.76 **	0.73 **	0.78 **	0.27 *	0.54 *	0.66 **	0.30 *	0.51 **	0.42 *	0.55 **	0.87 **
MD	−0.39 **	−0.39 **	−0.39 **	−0.16	−0.32	−0.28 *	−0.07	−0.20	−0.02	−0.22	−0.43 **
CL	0.42 **	0.20	0.30 *	0.27 *	0.33 *	0.19	0.19	0.26 *	0.27 *	0.42 **	0.40 **
CP	0.01	0.04	0.132	0.02	0.06	0.05	0.01	0.10	0.08	0.06	0.04
CS	−0.31 *	−0.38 *	−0.29 **	0.06	0.02	−0.29	0.34	−0.15	0.15	0.11	−0.29 **
**DCE-MRI Indices**											
BF	0.54	0.48	0.57 *	0.20	0.78 **	0.38	0.11	0.47 *	0.25	0.47 *	0.46 *
BV	0.36	0.35	0.53	0.15	0.76 **	0.31	0.11	0.22	0.22	046 *	0.49
k_ep_	0.41	0.50 *	0.53 *	0.01	0.32 *	0.38 *	0.13	0.30	0.24	0.47 *	0.48 *
PCI	−0.07	0.11	0.10	0.06	0.18	−0.03	0.17	−0.15	0.15	0.10	0.06

** Correlation is significant at the 0.01 level, * correlation is significant at 0.05 level. FA: fractional anisotropy, MD: mean diffusivity, CL: linear anisotropy, CP: planar anisotropy, CS: spherical isotropy, BV: blood volume, BF: blood flow, k ep: volume transfer constant, PCI: post-contrast signal intensity.

**Table 5 diagnostics-12-03041-t005:** Correlation between DTI and DCE indices with proinflammatory cytokines in the synovial fluid. Results are expressed as R values.

DTI-MRI Indices	Total Arthritis Patients (*n* = 80)		
	TNF-α	IL-1β	IL-6
FA	0.37 **	0.32 **	0.09
MD	−0.38 *	−0.86	−0.22
CL	0.47 *	0.28 *	0.13
CP	−0.16	−0.04	−0.14
CS	−0.06	−0.06	0.10
**DCE-MRI Indices**			
BF	0.32	0.12	0.22
BV	0.04	0.50	0.22
k_ep_	0.03	0.34	0.32
PCI	−0.05	0.27 *	0.02

** Correlation is significant at the 0.01 level, * correlation is significant at 0.05 level. FA: fractional anisotropy, MD: mean diffusivity, CL: linear anisotropy, CP: planar anisotropy, CS: spherical isotropy, BV: blood volume, BF: blood flow, k_ep_: volume transfer constant, PCI: post-contrast signal intensity.

**Table 6 diagnostics-12-03041-t006:** (a). Linear regression models (*n* = 80) with T lymphocyte population infiltration in the synovium. (b). Linear regression models (*n* = 80) with B lymphocyte and plasma cells infiltration in the synovium. (c). Linear regression models (*n* = 80) with macrophages and other inflammatory cytokines infiltration in the synovium.

**a**						
**IHC (T Cells)**	**DTI Metrices (Predictors)**	**β_0_**	**β_1_**	**R^2^**	**Sig.**	**SE (Estimates)**
CD 3	FA	−126.66	1269.34	0.64	<0.001	29.42
	MD	216.97	−38.06	0.15	0.001	45.02
	CL	108.49	798.55	0.20	<0.001	43.73
	CP	149.10	39.51	0.00	0.729	48.99
	CS	653.69	−663.50	0.09	0.01	46.66
	PCI	184.14	−0.01	0.00	0.58	48.91
CD 4	FA	−120.28	828.04	0.59	<0.001	21.02
	MD	104.73	−25.35	0.15	0.001	30.48
	CL	40.86	387.82	0.10	0.008	31.31
	CP	58.54	32.99	0.00	0.66	33.07
	CS	475.06	−547.63	0.14	0.002	30.69
	PCI	32.05	0.01	0.01	0.37	32.91
CD 8	FA	−56.01	494.02	0.73	<0.001	9.08
	MD	80.15	−16.30	0.21	<0.001	15.70
	CL	38.32	262.46	0.16	0.001	16.20
	CP	47.47	41.41	0.01	<0.001	17.62
	CS	261.52	−276.63	0.12	0.004	16.61
	PCI	38.18	0.00	0.01	0.41	17.67
**b**						
**IHC (B Cells)**	**DTI Metrices (Predictors)**	**β_0_**	**β_1_**	**R^2^**	**Sig.**	**SE (Estimates)**
CD 20	FA	11.50	125.45	0.00	0.02	13.52
	MD	43.24	−2.39	0.00	0.49	14.01
	CL	31.73	130.72	0.06	0.03	13.59
	CP	38.39	6.36	0.00	0.84	14.06
	CS	7.13	42.83	0.00	0.58	14.04
	PCI	31.2	0.00	0.00	0.59	14.04
CD138	FA	3.79	145.44	0.09	0.01	13.89
	MD	39.74	−2.26	0.00	0.53	14.56
	CL	28.20	135.19	0.06	0.03	14.11
	CP	35.30	5.16	0.00	0.87	14.60
	CS	−1.46	49.92	0.00	0.53	14.56
	PCI	13.98	0.01	0.031	0.15	14.37
**c**						
**IHC (Other Inflammatory Cells)**	**DTI Metrices (Predictors)**	**β_0_**	**β_1_**	**R^2^**	**Sig.**	**SE (Estimates)**
CD 68	FA	−7.32	768.64	0.46	<0.001	25.55
	MD	194.73	−19.34	0.08	0.020	33.46
	CL	152.87	177.52	0.02	0.26	34.54
	CP	153.13	68.23	0.01	0.40	34.69
	CS	568.98	−539.83	0.12	0.004	32.66
	PCI	173.92	−0.00	0.00	0.77	34.86
CD 54	FA	−12.44	219.97	0.26	<0.001	11.36
	MD	44.94	−5.27	0.04	0.10	12.96
	CL	31.58	81.98	0.03	0.17	13.04
	CP	33.36	20.25	0.00	0.51	13.19
	CS	−106.14	−92.85	0.02	0.20	13.07
	PCI	54.03	−0.01	0.02	0.21	13.08
IL-1β	FA	−48.16	357.25	0.36	<0.001	14.58
	MD	48.93	−10.94	0.09	0.013	17.41
	CL	11.56	335.76	0.26	<0.001	15.73
	CP	27.78	22.39	0.00	0.599	18.25
	CS	240.32	−278.34	0.11	0.005	17.16
	PCI	20.31	0.00	0.00	0.58	18.24
TNF-α	FA	−10.08	156.83	0.17	0.001	10.54
	MD	25.42	−0.44	0.00	0.87	11.60
	CL	18.94	99.11	0.05	0.050	11.27
	CP	21.95	18.61	0.00	0.49	11.56
	CS	82.60	−77.02	0.02	0.23	11.47
	PCI	9.17	0.00	0.02	0.21	11.46
Total cells infiltrating the synovium	FA	−213.27	3168.58	0.83	<0.001	43.67
	MD	637.21	−90.53	0.18	<0.001	96.76
	CL	385.53	1790.49	0.21	<0.001	95.00
	CP	468.73	141.85	0.00	0.56	106.95
	CS	1632.96	−1520.97	0.10	0.009	101.50
	PCI	493.40	−0.00	0.00	0.97	107.22

FA: fractional anisotropy, MD: mean diffusivity, CL: linear anisotropy, CP: planar anisotropy, CS: SPHERICAL isotropy, PCI: post-contrast signal intensity.

**Table 7 diagnostics-12-03041-t007:** Linear regression models (*n* = 80) with angiogenesis (CD34) in the synovium and dynamic contrast enhance magnetic resonance imaging variables.

Dependent Variables (IHC)	DCE Metrices (Predictors)	β_0_	β_1_	R^2^	Sig.	SE (Estimates)
CD 34	BF	16.83	0.32	0.66	<0.001	10.10
	BV	19.68	3.48	0.71	<0.001	9.22
	k_ep_	39.69	5.30	0.09	0.013	16.56
	PCI	0.01	29.71	0.02	0.214	17.20

BV: blood volume, BF: blood flow, k_ep_: volume transfer constant, PCI: post-contrast signal intensity.

## Data Availability

Data are available with the corresponding author and can be shared on reasonable request.

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
