# Peer review of "Diffusion Tensor and Dynamic Contrast-Enhanced Magnetic Resonance Imaging Correlate with Molecular Markers of Inflammation in the Synovium"

_diagnostics, 2022, doi:10.3390/diagnostics12123041_

Round 1
Reviewer 1 Report
This study examined direct correlation between cellular and molecular markers of synovial inflammation and MRI metrics. The DTI and DCE metrics can capture synovial inflammation in patients with chronic inflammatory arthritis and may become a non-invasive tool to assess degree of synovial inflammation. The study deisgn is novel, and manuscript is well-written with results and methods clearly presented. However, I have a few concerns:
1. What is the conventional MRI? It is better to be clear about the specific type of conventional MRI used in the abstract.
2. The authors used histological scoring system to classify patients into mild, moderate and severity, but they didn’t perform any subgroup analysis related to it. Investigating correlational patterns in different groups of patients based on severity can advance the understanding of current findings.
3. How is the sample size estimated? The authors may consider providing more detail on the calculation and formula.
4. Numerous correlations were examined in this study, which can lead to a very high risk of Type I error. Correction for multiple correlations should be performed to control for false positive results.
5. In Table 4, 5, S1 and S2, the authors should at least add a note that the values are correlation coefficient (r value) instead of significance level (p value) which can avoid misunderstanding.
6. It is better to keep at least 3 digits after the decimal point for the p value.
7. For the prediction utility, why did the authors look at the predictive ability of each single DTI or DCE metric? Establishing a multivariate linear regression model using all metrics or significant metrics to predict ICM parameters, evaluating the overall predictive performance and assessing which metric contributed most to the prediction is a better way to reach the highest prediction performance.
8. I didn’t find limitations mentioned in this study. At least, the correlation in each subgroup may be under-powered and not robust owing to the small sample size.
Author Response
This study examined direct correlation between cellular and molecular markers of synovial inflammation and MRI metrics. The DTI and DCE metrics can capture synovial inflammation in patients with chronic inflammatory arthritis and may become a non-invasive tool to assess degree of synovial inflammation. The study deisgn is novel, and manuscript is well-written with results and methods clearly presented. However, I have a few concerns:
- What is the conventional MRI? It is better to be clear about the specific type of conventional MRI used in the abstract.
Reply: We have added type of conventional MRI in the abstract.
- The authors used histological scoring system to classify patients into mild, moderate and severity, but they didn’t perform any subgroup analysis related to it. Investigating correlational patterns in different groups of patients based on severity can advance the understanding of current findings.
Reply: We thank the esteemed reviewer for pointing out our oversight on the correlation with histological scoring system. We have added a Supplementary table (S1).
- How is the sample size estimated? The authors may consider providing more detail on the calculation and formula.
Reply: We have added details of the sample size estimation in the methods section
- Numerous correlations were examined in this study, which can lead to a very high risk of Type I error. Correction for multiple correlations should be performed to control for false positive results.
Reply: Again we thank the esteemed reviewer for pointing it out. We have a=carried out Bonferroni correction and have mentioned this in the statistical analysis section.
- In Table 4, 5, S1 and S2, the authors should at least add a note that the values are correlation coefficient (r value) instead of significance level (p value) which can avoid misunderstanding.
Reply: Again we thank the reviewer for pointing this out. We have now added that results are expressed as R values.
- It is better to keep at least 3 digits after the decimal point for the p value.
Reply: We have mentioned p values in up to 3 places after decimal, wherever it is significant.
- For the prediction utility, why did the authors look at the predictive ability of each single DTI or DCE metric? Establishing a multivariate linear regression model using all metrics or significant metrics to predict ICM parameters, evaluating the overall predictive performance and assessing which metric contributed most to the prediction is a better way to reach the highest prediction performance.
Reply: We have carried out multivariate linear regression for parameters ; total number of cells infiltrating the synovium (Table S4) and CD34 (marker of angiogenesis) (Table S5).
- I didn’t find limitations mentioned in this study. At least, the correlation in each subgroup may be under-powered and not robust owing to the small sample size.
Reply: We have added limitation of our study (second last paragraph of the discussion) as has been rightly suggested by the esteemed reviewer.
Reviewer 2 Report
The manuscript is well written, well presented and contains novel idea. Further the manuscript is correct both scientifically and grammatically. It will help a lot to researchers working in the direction of diffusion tensors. Hence I recommend it to be accepted for publication in its current form.
Author Response
We thank the esteemed reviewer for his positive comments and recommendation of acceptance of the manuscript.
Round 2
Reviewer 1 Report
I have no further concerns, and this manuscript can be accepted in present form.